# The Effect of Short-Term Artificial Feed Domestication on the Expression of Oxidative-Stress-Related Genes and Antioxidant Capacity in the Liver and Gill Tissues of Mandarin Fish (*Siniperca chuatsi*)

**DOI:** 10.3390/genes15040487

**Published:** 2024-04-12

**Authors:** Zhou Zhang, Xiping Yuan, Hao Wu, Jinwei Gao, Jiayu Wu, Zhenzhen Xiong, Zhifeng Feng, Min Xie, Shaoming Li, Zhonggui Xie, Guoqing Zeng

**Affiliations:** Hunan Fisheries Science Institute, Changsha 410153, China; zz19961022@hotmail.com (Z.Z.); kerryuan@163.com (X.Y.); wh17380133463@163.com (H.W.); gaojinwei163@163.com (J.G.); wwjjyy1023@163.com (J.W.); 13763092902@163.com (Z.X.); fengzhifeng@sina.com (Z.F.); lishaoming1977@126.com (S.L.); xieice123@163.com (Z.X.); zengguoqing001@163.com (G.Z.)

**Keywords:** artificial feed domestication, oxidative stress, mandarin fish

## Abstract

To investigate whether Mandarin fish developed oxidative stress after being domesticated with artificial feed, we conducted a series of experiments. Oxidative stress is an important factor leading to diseases and aging in the body. The liver integrates functions such as digestion, metabolism, detoxification, coagulation, and immune regulation, while the gills are important respiratory organs that are sensitive to changes in the water environment. Therefore, we used the liver and gills of Mandarin fish as research materials. The aim of this study was to investigate the effects of short-term artificial feed domestication on the expression of oxidative stress genes and the changes in oxidative-stress-related enzyme activity in the liver and gills of Mandarin fish. We divided the Mandarin fish into two groups for treatment. The control group was fed with live bait continuously for 14 days, while the experimental group was fed with half artificial feed and half live bait from 0 to 7 days (T-7 d), followed by solely artificial feed from 7 to 14 days (T-14 d). The experimental results showed that there was no difference in the body weight, length, and standard growth rate of the Mandarin fish between the two groups of treatments; after two treatments, there were differences in the expression of genes related to oxidative stress in the gills (*keap1*, *kappa*, *gsta*, *gstt1*, *gstk1*, *SOD*, and *CAT)* and in the liver (*GPx*, *keap1*, *kappa*, *gsta*, *gstt1*, *gr*, and *SOD*). In the liver, GPx activity and the content of MDA were significantly upregulated after 7 days of domestication, while in the gills, SOD activity was significantly upregulated after 7 days of domestication and GPx activity was significantly downregulated after 14 days of domestication. These results suggest that artificial feed domestication is associated with oxidative stress. Moreover, these results provide experimental basic data for increasing the production of aquaculture feed for Mandarin fish.

## 1. Introduction

Aquaculture has become one of the fastest-growing agricultural and food production industries in the world [1]. Fish culturing is an important constituent part of aquaculture [2]. *S. chuatsi* (also known as Mandarin fish or Chinese perch) is an economically important fish cultured in China with high market value, and it belongs to the family Serranidae in the order Perciformes [3,4]. Mandarin fish is tender and easy to digest, without muscle spines, has a low cholesterol content and extremely high nutritional value, and is loved by consumers. Mandarin fish have a unique diet, requiring live fish or shrimp as bait during the mouth-open period, which they feed on for their entire life [5]. The large-scale artificial breeding of carnivorous fish is extremely difficult. Since the late 1980s, certain progress has been made with regard to the domestication technology, feed nutrition and formula, and feed breeding research of *S. chuatsi* [6]. Liang [7] and Wu [8] addressed the problem of active compound feeding of *S. chuatsi* by studying their feeding behavior and sensation mechanisms. With continuous advancements in replacing live bait fish or fresh ice fish with artificial feed, the impact of feed on the growth performance and nutritional value of carnivorous fish has received increasing attention [9]. Studies have shown that, compared to live-baited *S. chuatsi*, the conversion rate of *S. chuatsi* fed with artificial feed is higher, the fullness and viscera-to-body ratio are significantly lower, and the meat quality and taste are better [10]. The content of saturated fatty acids, monounsaturated fatty acids, and polyunsaturated fatty acids in the muscles of *S. chuatsi* receiving artificial feed is significantly higher than that in the live bait group, indicating that artificial feed can provide higher-quality fatty acids through a balanced nutrient supply [11]. The gut microbiota’s α-diversity index in the *S. chuatsi* receiving artificial feed is significantly higher than that of live-baited *S. chuatsi* [12]. However, during our domestication process, we found that the survival rate of *S. chuatsi* receiving artificial feed was still much lower than that of live-baited *S. chuatsi*. To explore the reasons for this phenomenon, we conducted many experiments. In this study, we mainly investigated the impact of feed domestication on the expression of oxidative stress-related genes in *S. chuatsi*, thereby affecting their survival rate. There is currently limited research on the effects of artificial feed on the body of Mandarin fish. In order to further improve the practical level of artificial feed, it is necessary to study the effects of domestication of Mandarin fish with artificial feed on the body.

With the development of large-scale and intensive aquaculture, excessive aquaculture density and poor water quality and environment have led to an increase in stress factors in the environment, resulting in oxidative stress in farmed animals [13]. Oxidative stress is a common bodily response mechanism that occurs after being affected by harmful factors and is related to the occurrence of many diseases [14]. Oxidative stress is particularly common in aquaculture. When the fish body is strongly stimulated, it can cause the antioxidant system in the body to lose balance, leading to metabolic disorders. Long-term oxidative stress can affect the growth rate of fish and lead to a decline in their quality [15]. Previous studies have suggested that excessive dietary fat levels can lead to a decline in the growth performance of young carp, affecting their body composition, blood lipid content, and glycolysis, resulting in oxidative stress [16]. Animal organisms have formed a set of antioxidant systems during the evolutionary process, including enzymatic and nonenzymatic systems. Among them, the enzymatic antioxidant system includes SOD, CAT, and so on [17]. Superoxide dismutase, catalase, and glutathione peroxidase are important antioxidant enzymes that can clear reactive oxygen species in the body to maintain stable levels of intracellular hydrogen peroxide and superoxide anions [18]. Glutathione is a nonenzymatic antioxidant that can serve as a substrate or cofactor for glutathione transferase and glutathione peroxidase. The three coordinate and participate in bodily defense, rapidly clearing oxygen free radicals and peroxides in cells. Glutathione plays an important protective role in maintaining cellular redox balance, normal cell growth and development, and homeostasis [19,20]. Malondialdehyde (MDA) is one of the final products of lipid peroxidation, which can oxidize the SH group of proteins, as well as crosslink sugars, lipids, nucleic acids, etc. Its content can indirectly reflect the level of oxygen free radicals and lipid peroxidation in the body and the degree of cellular oxidative damage [21]. When the content of MDA accumulates excessively, it can lead to cell toxicity [17]. In this study, we conducted experiments on the expression and enzyme activity changes of oxidative stress-related genes in the liver and gills of Mandarin fish after domestication with artificial feed, providing basic experimental data for increasing the production of aquaculture feed for Mandarin fish.

## 2. Materials and Methods

### 2.1. Ethics Statement

All animal procedures were strictly carried out in accordance with the relevant guidelines (license No. HNFI20230610) and were approved by the Animal Welfare and Ethics Committee of Hunan Fisheries Science Institute (program approval on 22 December 2022).

### 2.2. Animal Holding and Experimental Design

Healthy *S. chuatsi* (*n* = 5000; mass: 4.04 ± 0.35 g; total length: 6.53 ± 0.54 cm) were sourced from the same batch of the Hunan Fisheries Science Institute (Changsha, China). Both the experimental group and control group were divided into three replicates, with 5000 fish raised in each replicate tank. All fish were reared in factory aquaculture tanks with a diameter of 4 m and a depth of 2 m, which were sterilized before use. The upper parts of the tanks were cylindrical, and the lower parts were conical. The water temperature was kept at 26 ± 2 °C, dissolved oxygen was ≥5 mg/L, pH was 7.2–7.5, and the adaptation period was one week. During the adaptation period, all fish were fed with live bait fish (*Mrigal carp*) twice per day to clear satiety, with approximately 80 live bait fish used per kilogram, and a feeding rate of 20% wet weight. After this period, the fish were divided into an experimental group and a control group. The control group continued to receive live bait fish, while the experimental group was fed artificial feed and live fish in a 1:1 ratio for 7 days, followed by solely artificial feed for 7–14 days (the formula for artificial feed is crude protein ≥ 48; coarse fat ≥ 7; coarse fiber ≥ 3; calcium 1.0–5.0; total phosphorus 1.2–3.0; lysine ≥ 2.9, Guangdong Shangshang Biotechnology Co., Ltd., Zhanjiang, China). The ingredients of the artificial feed formula included fish meal, krill meal, phospholipids, fish oil, soybean oil, flour, vitamins, mineral elements, etc. The control group and experimental group were each sampled at 7 and 14 days. Before sampling, both groups of fish were starved for 24 h, and then 5 fish from each group were randomly selected from the two tanks. During the sampling process, the selected fish were first anesthetized with MS-222 (Sigma Aldrich, St. Louis, MO, USA), then they were weighed, and their body length was measured. The fish were dissected, and the liver and gills were stored separately in a −80 °C freezer in Eppendorf tubes for subsequent use.

### 2.3. Antioxidant Capacity Assays

In the gill and liver samples from different groups of Mandarin fish, the activity levels of superoxide dismutase (SOD), catalase (CAT), and glutathione peroxidase (GPx) were measured, as well as the levels of malondialdehyde (MDA). The standard kits obtained from Shanghai ZCIBIO Technology Co. Ltd., Shanghai, China, were used for this test, following the instructions provided by the manufacturer. The catalog numbers were No. ZC-S0350, No. ZC-S0351, No. ZC-S0860, and No. ZC-S0343. The enzyme assay temperature was 25 °C. All analyses were conducted in triplicate.

### 2.4. Gene Expression Analysis

In order to extract total RNA from the liver and gill samples of Mandarin fish, we used the RNA extraction kit from FOREGENE (Cat. No. RE-03014, Chengdu, China). We used two methods to test the extracted RNA to ensure its usability. Firstly, the concentration and quality of the RNA were calculated using a NanoDrop 2000 spectrophotometer (NanoDrop Technologies, Wilmington, DE, USA). When the ratio of A260:280 was close to 1.9, we determined that the purity of the RNA was good. Secondly, gel electrophoresis was used to detect the brightness and singularity of the band. When the band was single and bright, we determined that RNA was available. It should be noted that RNA is easily degraded, and during the extraction and testing process, the environment and reagent should be controlled as much as possible to ensure the absence of RNase. In order to conduct subsequent experiments, RNA was reverse-transcribed into cDNA. Firstly, the RNA was diluted to a concentration of 15 ng/μL. Then, according to the manufacturer’s instructions, the PrimeScript ™ RT reagent and gDNA eraser kit (Cat No.RE-03014, FOREGENE, Chengdu, China) were used to synthesize cDNA using 2 mcg of RNA as templates. Subsequently, real-time fluorescence quantitative PCR was carried out. The reaction system consisted of 2 µL of cDNA template, 0.8 µmol/L of each primer, 6.4 µL of water, and 10 µL 2 × SYBR Green Master Mix kit (Takara, Japan). The experimental instrument was a LightCycler^®^ 480 system from Roche, Switzerland. The primer sequences of the target genes were sourced from the NCBI, and synthesized by Sangon Biotech Co., Ltd. (Shanghai, China). The sequences of primers can be found in Table 1. The usability of each pair of primers was tested with PCR and gel electrophoresis before use. The rpl13 gene [22] was employed as a reference gene in this experiment to determine the relative mRNA expression of the target genes using the 2-∆∆Ct method. (CAT, catalase; GPx, glutathione peroxidase; GR, glutathione reductase; GSTA, glutathione S-transferase-α; GSTK1, glutathione S-transferase kappa 1; GSTT1, glutathione S-transferase theta 1b; keap1, kelch-like ECH associated protein 1; SOD, superoxide dismutase; rpl13, ribosomal protein L13).

### 2.5. Statistical Analysis

All of the analyses were conducted using SPSS 22.0 statistical software (IBM, Chicago, IL, USA). The data are presented as the mean ± standard deviation (SD). The data follow a normal distribution, and the variances of each sample are uniform. The homogeneity of variance was assessed using Levene’s equal variance test, while the normal distribution was examined using the Shapiro–Wilk test for all data. The comparison between two sets of data was conducted using a *t*-test, and a *p*-value of <0.05 was defined as indicating a significant difference. Different numbers of asterisks represent the significance of differences. (*: *p*-value < 0.05; **: *p*-value < 0.01; ***: *p*-value < 0.001; ****: *p*-value < 0.0001).

## 3. Results

### 3.1. Artificial Feed Domestication Did Not Affect Weight and Length in S. chuatsi

After 14 days of the feeding experiment, as shown in Figure 1, no significant difference was observed in the body length (BL) or body weight (BW) of the *S. chuatsi*. There was also basically no difference in growth between domesticated *Siniperca chuatsi* fed with artificial feed and those fed with live bait in the short term. 

### 3.2. Artificial Feed Domestication Changed the mRNA Expression of Gill Antioxidant Genes in S. chuatsi

As shown in Figure 2, the relative expression of the *GPx*, *keap1*, *kappa*, *gr*, *gsta*, *gstt1*, *gstk1*, *SOD*, and *CAT* genes was detected in the gills of *S. chuatsi*. The results showed that there were no significant differences in the relative expression of *GPx*, *gr*, and *gstk1* between the control group and the experimental group. The expression of *keap1*, *kappa*, *gsta*, *SOD*, and *CAT* was significantly downregulated after 14 days, and the expression of *kappa* was significantly downregulated after 7 days, while the expression of *gstt1* and *gstk1* was significantly upregulated after 7 days.

### 3.3. Artificial Feed Domestication Changed the Antioxidant Ability of Gills in S. chuatsi

To study whether Mandarin fish experienced oxidative stress after domestication, the gill tissues were investigated for antioxidant molecular activity using four different indicators (Figure 3A–D). The results demonstrated that the SOD activity in the gill tissue was significantly lower in the 7-day experimental group than it was in the control group (Figure 3A), while CAT activity was apparently unchanged (Figure 3B) after 14 days. After domestication, the GPx activity was significantly lower than in the control group (Figure 3C), while the content of MDA was apparently unchanged (Figure 3D).

### 3.4. Artificial Feed Domestication Changed the mRNA Expression of Liver Antioxidant Genes in S. chuatsi

As shown in Figure 4, the relative expression of *GPx*, *keap1*, *kappa*, *gr*, *gsta*, *gstt1*, *gstk1*, *SOD*, and *CAT* genes were detected in the livers of *S. chuatsi*. The results showed that there were no significant differences in *gstk1* and *CAT* relative expression between the control group and experimental group. The expression of *GPx*, *keap1*, *kappa*, *gr*, *gsta*, and *gstt1* was significantly downregulated after 14 days, and the expression of *gr* and *SOD* was significantly upregulated after 7 days.

### 3.5. Artificial Feed Domestication Changed Liver Antioxidant Ability in S. chuatsi

To study whether Mandarin fish experienced oxidative stress after domestication, the liver tissues were investigated for antioxidant molecular activity using four different indicators (Figure 5A–D). The activity of SOD and CAT was apparently unchanged (Figure 5A,B). The GPx activity was significantly higher in the 7-day experimental group than it was in the control group (Figure 5C), and the MDA content was significantly higher in the 7-day experimental group than it was in the control group (Figure 5D).

## 4. Discussion

Feeding is the only way for fish to obtain energy for various activities such as growth, development, and reproduction [23]. Research has shown that nutritional imbalance and anti-nutritional factors in compound feed can lead to an increase in liver fat content and a decrease in food intake in Mandarin fish [24]. Our research indicates that after short-term domestication with feed, there was no difference in the body weight, length, and standard growth rate of Mandarin fish compared to live fish fed with artificial feed. We assume that this may be due to the short domestication time, resulting in insignificant differences. *Micropterus salmoides* is a carnivorous fish that prefers live bait, too. Studies have shown that the timing of domestication and the nutritional composition of the artificial feed are key factors affecting the survival rate of *M. salmoides* [25]. In the future, we need to strengthen our understanding of the nutritional requirements and digestive system development of Mandarin fish, which is the key to improving the survival rate.

Reactive oxygen species (ROS) in organisms are normally maintained in a dynamic equilibrium with the contribution of the antioxidant scavenging systems, whereas nutritional imbalances could disrupt the homeostasis of the pro-oxidant–antioxidant system, leading to oxidative stress [26,27]. The antioxidant defense system in the body plays an important role in protecting the body from oxidative damage. In order to maintain normal physiological functions of the body, the antioxidant defense system strictly controls the amount of ROS by preventing or delaying unnecessary oxidation of related substrates, ensuring the maintenance of the “redox homeostasis” in the body [28,29]. Fish have evolved antioxidant defense mechanisms to counteract stress, particularly through Nrf2-Keap1 signaling [30]. Nrf2-Keap1 signaling is an evolutionarily conserved intracellular defense mechanism to counteract oxidative stress by regulating the activities of antioxidant enzymes [31]. The antioxidant enzyme SOD converts free radicals to oxygen and hydrogen peroxide, which is then catalyzed by CAT to form water and oxygen [32]. The Nrf2-Keap1 system is an upstream factor that initially perceives and generates commands to protect cells from oxidative stress-induced damage [33]. In this study, we found that after 7 days of domestication, the expression of *SOD* and *gr* mRNA was significantly upregulated in the liver. While the expression of *keap1*, *kappa*, *gsta*, *CAT*, and *SOD* mRNA was significantly downregulated in gills after 14 days of domestication. Similar results were observed in gene expression in the liver. After 7 days of domestication, the expression of oxidative stress related genes in Mandarin fish was generally upregulated. We assume that this is due to the Mandarin fish increasing the expression of oxidative stress genes to enhance antioxidant capacity. However, after 14 days of domestication, the expression of oxidative-stress-related genes was generally downregulated. We believe this may be due to oxidative damage and decreased gene expression in the Mandarin fish. This indicates that Mandarin fish experience oxidative stress during short-term domestication and resist oxidative damage by regulating gene expression.

MDA is a biomarker that reflects the level of oxidative stress in the body, and its content can be used to evaluate levels of tissue damage [34]. In this study, there was a significant difference in MDA content in the liver after 7 days of domestication, but there was no difference in MDA content after 14 days, indicating that the liver experienced oxidative stress during the short-term domestication of Mandarin fish. However, health status may be improved by eliminating excess ROS through the body’s antioxidant defense system, as ROS leads to the accumulation of MDA and other lipid peroxides through the decomposition products of hydrogen peroxide. SOD and GPx are fundamental components of the enzymatic antioxidant system. SOD can convert O_2_^−^ to H_2_O_2_, while GPx catalyzes the conversion of glutathione hydrogen peroxide to H_2_O [35,36,37]. In this study, the GPx content in the liver significantly increased after 7 days of domestication in the case of Mandarin fish, but there was no difference after 14 days, indicating that the antioxidant capacity of GPx enzymes may play a major role within 7 days of domestication; the GPx content of gills significantly decreased after 14 days of domestication.

Artificial feed can cause varying degrees of oxidative stress in aquatic animals. Research by Cong has shown that compared to the fresh Pacific jade tendon fish (*Ammodytes personatus*), using compound feed can lead to a decrease in the antioxidant capacity of the pearl gentian grouper (*Epinephelus lanceolatus*) [38]. Different types of feed can, to some extent, alter the antioxidant capacity of aquatic animals. Research has found that adding 47% carbohydrates to the feed of *Megalobrama ambrycyphala* significantly increases GOT activity in its serum, while liver SOD and T-AOC activities show a decreasing trend [39]. In a study of *M. salmoides*, it was found that with an increase in dietary starch levels, the MDA content in the liver increased [40]. High sugar feed can cause oxidative stress in Yellow River carp, exacerbating the degree of lipid peroxidation in the body, thus damaging health [41]. The liver SOD, CAT, GPx, GR, and T-AOC activity of black carp, Yellow River carp [42], and spotted seabass [43] fed a high-fat diet were significantly reduced, while the MDA content was significantly increased [44]. Research has shown that adding baicalein to feed can reduce the serum MDA content of grass carp (*Ctenopharyngodon idella*) while increasing CAT and SOD activities, thereby enhancing antioxidant capacity [45]. Adding quercetin to feed can increase the antioxidant enzyme activity in the liver of Yellow River carp [46], zebrafish [47], common carp (*Cyprinus carpio*) [48], and snakehead (*Channa argus*) [49] and reduce liver MDA content. Adding an appropriate amount of Lycium barbarum Polysaccharide to feed can not only improve the growth performance and feed utilization of snakehead fish but also enhance its antioxidant capacity. In addition, it can also enhance its immune system [50]. Studies have shown that adding phospholipids to feed can significantly reduce the content of MDA in the liver of juvenile snakehead fish, improve total antioxidant capacity and SOD activity, and alleviate oxidative damage to the liver of juvenile snakehead fish. This indicates that supplementing phospholipids in feed helps to enhance the antioxidant capacity of juvenile snakehead fish and reduce lipid peroxidation [51]. These provide ideas for improving artificial feed to domesticate Mandarin fish and enhance their antioxidant capacity in the future.

## 5. Conclusions

The expression of oxidative stress genes in the liver (*GPx*, *keap1*, *kappa*, *gr*, *gsta*, *gstt1*, *SOD*, and *gr*) and gills (*keap1*, *kappa*, *gsta*, *gstt1*, *gstk1*, *SOD*, and *CAT*) of Mandarin fish domesticated with artificial feed changed, and their antioxidant capacity decreased. These results suggest that artificial feed domestication is associated with oxidative stress. We assume that domesticated mandarin fish are more susceptible and have a lower survival rate, possibly due to oxidative stress and decreased antioxidant capacity. In the future, we can focus on how to reduce the oxidative stress that Mandarin fish experience during domestication with feed, for example, by adding antioxidants to feed. Moreover, these results provide basic experimental data for increasing the production of aquaculture feed for Mandarin fish.

## Figures and Tables

**Figure 1 genes-15-00487-f001:**
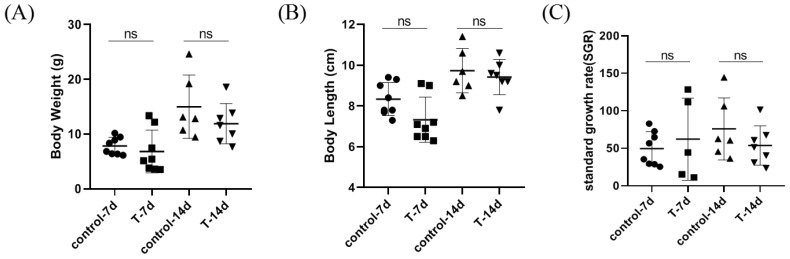
The figure shows the growth performance of different feed in *S. chuatsi.* (**A**) Body weight; (**B**) body length; and (**C**) standard growth rate. Note: ns represents *p* > 0.05; ●, ■, ▲ and ▼ represent different repetitive samples in each group, respectively.

**Figure 2 genes-15-00487-f002:**
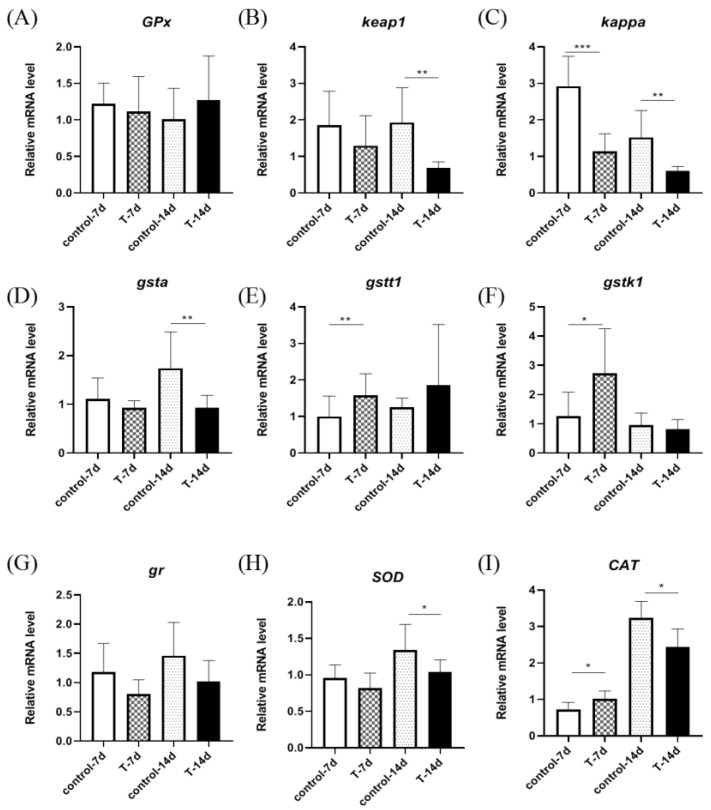
The figure shows the xpression of gill antioxidant genes in *S. chuatsi* after 7 days and 14 days of artificial feed. (**A**) *GPx* l; (**B**) *keap1*; (**C**) *kappa*; (**D**) *gsta*; (**E**) *gstt1*; (**F**) *gstk1*; (**G**) *gr* (**H**) *SOD*; (**I**) *CAT*. Note: *t*-test; *, **, and *** represent *p* < 0.05, *p* < 0.01, and *p* < 0.001, respectively.

**Figure 3 genes-15-00487-f003:**
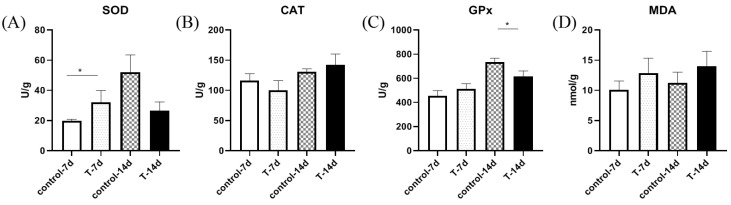
The figure shows artificial feed domestication changed gill antioxidant ability in *S. chuatsi*. The activity of SOD (**A**), CAT (**B**), and GPx (**C**), and the content of MDA (**D**) in each fish after feeding with artificial feed or live bait fish. Note: * represents *p* < 0.05.

**Figure 4 genes-15-00487-f004:**
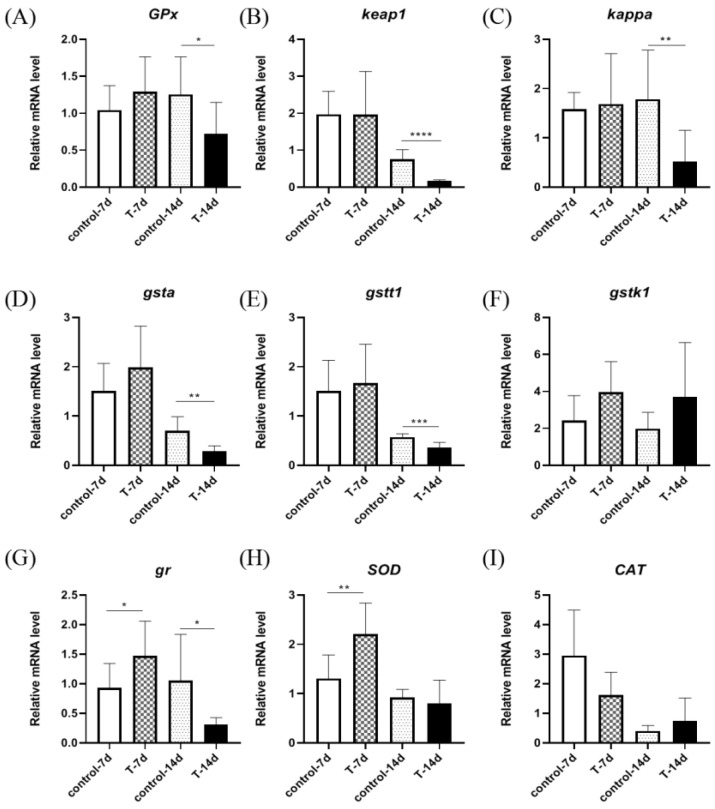
The figure shows the expression of liver antioxidant genes in *S. chuatsi* after 7 days and 14 days of artificial feed. (**A**) *GPx*; (**B**) *keap1* i; (**C**) *kappa*; (**D**) *gsta*; (**E**) *gstt1*; (**F**) *gstk1*; (**G**) *gr*; (**H**) *SOD*; (**I**) *CAT*. Note: *t*-test; *, **, ***, and **** represent *p* < 0.05, *p* < 0.01, *p* < 0.001, and *p* < 0.0001, respectively.

**Figure 5 genes-15-00487-f005:**
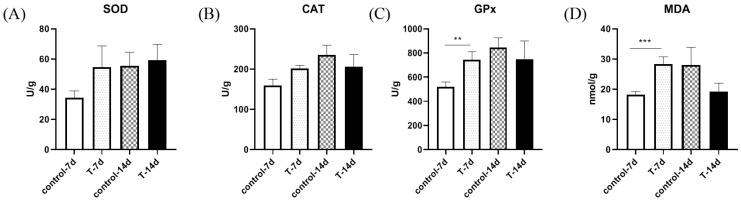
The figure shows that artificial feed domestication changed liver antioxidant abilities in *S. chuatsi*. The activity of SOD (**A**), CAT (**B**), and GPx (**C**) and the content of MDA (**D**) in each fish after feeding with artificial feed or live bait fish. Note: ** and *** represent *p* < 0.01 and *p* < 0.001, respectively.

**Table 1 genes-15-00487-t001:** The table shows the qPCR primers of Mandarin fish.

Primer	Primer Sequence (5′-3′)	Function
CAT	F: CCCGATATGGTGTGGGACTT R: GAAGGTGTGAGAGCCGTAGC	Antioxidant-related
GPx	F: GCCCATCCCCTGTTTGTG R: AACTTCCTGCTGTAACGCTTG	Antioxidant-related
GR	F: CAGGCATCCTTTCCACCC R: TCCAGTCCTCTGTCCGTTTTA	Antioxidant-related
GSTA	F: TGGAGCACAAGTCACAGGAAG R: TGCTGCGTAGGATTCATTCA	Antioxidant-related
GSTK1	F: AAGCCTCCTGGTCTGGTTCC R: ACCCGCTCCACCTGCTTG	Antioxidant-related
GSTT1	F: CGAAGGCGAAGATGGACG R: GATTTTGTCGCCGATGATGAA	Antioxidant-related
KAPPA	F: GTGGCAACCCAGGAGGAG R: GGGAATGGCAACGGACA	Antioxidant-related
KEAP1	F: TTCCACGCCCTCCTCAA R: TGTACCCTCCCGTATG	Antioxidant-related
SOD	F: ACAATCCCCACGGCAAGAAT R: TTGAGTAGGGGCCAGTGAGG	Antioxidant-related
Rpl13	F: CACAAGAAGGAGAAGGCTCGGGT R: TTTGGCTCTCTTGGCACGGAT	Housekeeping gene

## Data Availability

The study data are available upon request from the corresponding author.

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
