# Peer review of "The Effect of Short-Term Artificial Feed Domestication on the Expression of Oxidative-Stress-Related Genes and Antioxidant Capacity in the Liver and Gill Tissues of Mandarin Fish (Siniperca chuatsi)"

_genes, 2024, doi:10.3390/genes15040487_

Round 1
Reviewer 1 Report
Comments and Suggestions for Authors
The topic of the study is original and of interest for fish culture and for understanding the feeding physiology of fish.
The study has 3 weaknesses:
The duration of the experiment is short. Differences in feeding regimes (in particular in growth and mortality) might become evident after prolonged feeding times due to malnutrition. It is clear that the experiment cannot be changed anymore. Maybe the authors can address this problem in more details in discussion. Also comparative data on duration of feeding experiments in warm water species could be useful.
The experiment needs better description in aspects of the used aquaculture system (RAS/flow through), stocking density, tank repetitions. Experiment must also be monitored in aspect of water quality, fish health, mortality rates.
The discussion needs more details about the functions of genes, the differences in gene expression over time and between control and treatment. As written, discussion is mainly limited to interpretation of results on keap1
Examples for points which might be worked out: According to charts liver gsta and gstt1 decrease in control with time (from 7-14d) – how is this interpreted. In gills only gsta decreased while gstt1 seems constant. Other point which should be considered: There exist also differences in results on gene expression and enzyme activity: E.g. while cat and SOD genes are down regulated, enzyme activity is similar (Fig. 5). Also this point is worth to be discussed. A more detailed discussion of results could improve the impact of the manuscript very significantly.
Specific comments:
Line 18,19: The gills are important organs for material exchange – please define
Line 20/21: It should be considered, that all fish kept in tanks are under conditions of domestication. The type of food is only one of many artificial factors interacting with fish.
Line 95: Which live bait fish species was offered? Please define.
Line 97/98: The term family is unclear: from the same parents / strains?
Lines 109/110: For determination of growth rates n = 5 is a very low sample number, as fish do not grow uniformly
Lines 119-123: Activity of enzymes depend on the assay temperature. This should be defined.
Table 1: The meanings/functions of the analyzed genes should be defined.
In the diagrams only variables from the same date were statistically compared with each other. I recommend to compare the variables also over time as this is important information ( 7d versus 14 d)
The result text should be worked out in more detail. The description in the text is not always conform to changes shown in the diagrams.
Author Response
Thank you for review very much. We are very grateful for you reviewing our manuscript and providing the valuable comments.
comment
The duration of the experiment is short. Differences in feeding regimes (in particular in growth and mortality) might become evident after prolonged feeding times due to malnutrition. It is clear that the experiment cannot be changed anymore. Maybe the authors can address this problem in more details in discussion. Also comparative data on duration of feeding experiments in warm water species could be useful.
Response
Thank you for your advice. In our early culturing work, we found that feed domesticated mandarin fish are more susceptible and have a lower survival rate than live bait mandarin fish. Micropterus salmoides is a carnivorous fish that prefers live bait, too. Studies have shown that the timing of domestication and the nutritional composition of the artificial feed are key factors affecting the survival rate of Micropterus salmoides [1]. In the future, we need to strengthen our understanding of the nutritional requirements and digestive system development of Mandarin fish, which is the key to improving the survival rate of seedlings. In our revised manuscript, we discussed more details about this problem.
[1] Liu, W.; Hong, Y.C.; Sun, K.H.; Chai, Y. Research progress on feeding habits and domesticated feed combinations of Micropterus salmoides fry. Nutrition and Feed. 2022, 8, 68-69. DOI: 10.14184/j.cnki.issn1004-843x.2022.08.025
comment
The experiment needs better description in aspects of the used aquaculture system (RAS/flow through), stocking density, tank repetitions. Experiment must also be monitored in aspect of water quality, fish health, mortality rates.
Response
Thank you for your review. The aquaculture system we used was flow through, the stocking density was 250 fish per cubic meter of water, tank repetitions were 2. The water temperature was kept at 26 ℃± 2 ℃, dissolved oxygen was ≥5 mg/L, pH was 7.2-7.5. Due to the short duration of the domestication experiment, we didn’t detect large-scale diseases, and the fish remained healthy. Moreover, due to the large number of fish we domesticate, the mortality rate was much lower than 1%, which could be relatively negligible.
comment
The discussion needs more details about the functions of genes, the differences in gene expression over time and between control and treatment. As written, discussion is mainly limited to interpretation of results on keap1.
Response
Thanks for your review. In our revised manuscript, we added more details about the functions of genes, the differences in gene expression over time and between control and treatment.
Comment
Examples for points which might be worked out: According to charts liver gsta and gstt1 decrease in control with time (from 7-14d) – how is this interpreted. In gills only gsta decreased while gstt1 seems constant. Other point which should be considered: There exist also differences in results on gene expression and enzyme activity: E.g. while cat and SOD genes are down regulated, enzyme activity is similar (Fig. 5). Also this point is worth to be discussed. A more detailed discussion of results could improve the impact of the manuscript very significantly.
Response
Thanks for your review. We didn’t compare the changes in gene expression levels at different times because there were too many variables in Mandarin fish at different times. Gsta and gstt1 respectively control the synthesis of glutathione S-transferase-α and glutathione S-transferase theta 1b. There were differences in the results of gene expression and enzyme activity, and we have also found the same phenomenon in other studies on feeding Mandarin fish to starvation [2]. We assume that there are two possibilities. The first possibility is that there may be experimental errors. The second possibility is the equation of time between transcription and translation, which leads to a time difference between gene expression and enzyme activity.
[2] Wu, P.; Chen, L.; Cheng, J.; Pan, Y.X.; Zhu, X.; Chu, W.Y.; Zhang, J.S. Effect of starvation and refeeding on reactive oxygen species, autophagy and oxidative stress in Chinese perch (Siniperca chuatsi) muscle growth. Fish Biology. 2022, 101: 168-178. doi: 10.1111/jfb.15081
Comment
Line 18,19: The gills are important organs for material exchange – please define
Response
Thank you for your review. We are sorry for our incorrect description. Gills are important respiratory organs that are sensitive to changes in the water environment [3]. We have revised the description in our manuscript.
[3] Panserat, S.; Hortopan, G.A.; Plagnes-juan, E. Differential gene expression after total replacement of dietary fish meal and fish oil by plant products in rainbow trout (Oncorhynchus mykiss) liver. Aquaculture, 2009, 294(1-2): 123-131.
Comment
Line 20/21: It should be considered, that all fish kept in tanks are under conditions of domestication. The type of food is only one of many artificial factors interacting with fish.
Response
Thank you for your recommendation. The two groups of fish are kept in different tanks but the same pond, with consistent environmental factors such as breeding density, feeding time, and dissolved oxygen, except for different types of feed.
Comment
Line 95: Which live bait fish species was offered? Please define.
Response
Thank you for your review. We used Mrigal carp as live bait fish, which were sourced from Hunan Fisheries Science Institute. And we have added this in the manuscript.
Comment
Line 97/98: The term family is unclear: from the same parents / strains?
Response
Thank you for your review. The experimental fish were from the same strains, we have revised in the manuscript.
Comment
Lines 109/110: For determination of growth rates n = 5 is a very low sample number, as fish do not grow uniformly
Response
Thank you for your review. We apologize that number of fish were randomly selected for growth measurement in this experiment. We sampled 6-8 samples, but the sample amount was still not enough. For this suggestion, we will increase the number of samples for determining the growth rate of fish in future experiments.
Comment
Lines 119-123: Activity of enzymes depend on the assay temperature. This should be defined.
Response
Thanks for your comment and reminder. The assay temperature was 25℃ of the enzymes. We have added this point in our revised manuscript.
Comment
Table 1: The meanings/functions of the analyzed genes should be defined.
Response
Thank you for your review. We have added the functions of the analyzed genes in Table 1.
Comment
In the diagrams only variables from the same date were statistically compared with each other. I recommend to compare the variables also over time as this is important information (7d versus 14 d)
Response
Thank you for your suggestion. We didn’t compare the variables over time because our fish are kept in factory captive tanks, there may be slight fluctuations in environmental factors such as dissolved oxygen and water temperature on different days. We believe that this comparison may not be able to achieve a single variable, so we did not compare them at different times.
Comment
The result text should be worked out in more detail. The description in the text is not always conform to changes shown in the diagrams.
Response
Thank you for your carefully review. We’re sorry for our carelessness. We have checked our results section, and revised more details in our manuscript.
Reviewer 2 Report
Comments and Suggestions for Authors
In this manuscript, the authors report the results of their experiment concerning the effects of short-term artificial feed domestication on enzyme antioxidant activities and the expression of oxidative stress genes in the liver and gills of Mandarin fish. Firstly, English needs a thorough overhaul. There are "non-sense" sentences and many grammatical or typo errors. Furthermore, words and expressions not normally used in aquaculture appear, as if the authors were unfamiliar with this activity. Moreover, they compare two different feeding approaches (live baits vs live baits + artificial feed), but they do not indicate the quality, or crude composition of the live bait, and the constituents (raw materials) of the artificial feed. In this way, it is not easy to understand the reason for the observed differences. Furthermore, it would have been appropriate to carry out the analyses at the beginning of the experiment, before diversifying the feeding. Below, I report minor/major suggestions and requests for clarification to improve the manuscript.
Abstract
The first part (lines 3-21) is only descriptive, therefore it must be reduced to essential information.
Introduction
Line 41. The reference 1 does not fit with the topic. It is necessary to indicate a more recent and general paper, such as DOI: 10.1111/jwas.12963
Materials and Methods
Line 98. Batch is better than family. Do barrels mean tanks? Please replace.
Lines 103-107. It is not described how many replicates for each experimental condition and how many fish for each replicate.
Lines 105-107. Here the formula of the artificial feed is reported. But what are the constituents (raw materials) and their percentages? It would probably be more appropriate to report this information in a table.
Line 101. “live bait fish”. More information is needed regarding the species, size, method of administration, density, method of distribution, etc. Have you evaluated the quality of live bait fish? Have you at least compared the energy balance between the two food theses?
Lines 119-123. For commercial kits, the codes must be indicated.
Line 120. MDA is not an antioxidant assay.
Line 153-150. According to my opinion, the correct statistical analysis is the two-way analysis of variance (ANOVA), with diet and days as factors, also considering their interaction.
Results
Lines 153-157 and Fig. 1. It is incorrect to represent growth as a weight gain but a standard growth rate (SGR). Moreover, in line 145 “The data are presented as mean ± standard deviation” is reported, whereas in Fig. 1 the data are presented as mean, SD, percentiles, and maximum and minimum values (maybe).
Figs. 3 and 5. Other symbols appear in the bars; what do they represent, median, percentiles or what else?
Discussion
The discussion needs to be reshaped. In some parts, information is indicated that is not always connected and discussed in light of the results obtained from this research.
Line 262. Reference 44 is not reported.
Conclusions
The conclusions are similar to a short summary. They must be rewritten and mainly indicate what, in the light of the results obtained, are the prospects for breeding of this species and any new research on the merits.
Comments on the Quality of English LanguageEnglish needs a thorough overhaul. There are "non-sense" sentences and many grammatical or typo errors.
Author Response
Thank you for review very much. We are very grateful for you reviewing our manuscript and providing the valuable comments.
Comment
The first part (lines 3-21) is only descriptive, therefore it must be reduced to essential information.
Response
Thank you for your recommendation, we have reduced the part to essential information in our revised manuscript.
Comment
Line 41. The reference 1 does not fit with the topic. It is necessary to indicate a more recent and general paper, such as DOI: 10.1111/jwas.12963
Response
Thank you for your comment and recommended reference for us. We have read this paper carefully. This paper showed that two dietary betaine supplementations on growth, feed utilization, lipid metabolism, and immune response of white shrimp (Litopenaeus vannamei) fed two levels of compound attractants. But this article is not very suitable to be placed at the beginning of the introduction section, and there will be no continuity with the following text. Our reference 1 and subsequent reference 2 have continuity. However, we will focus on research on immune response in the future, it will be very helpful for us to carry out related research in the future.
Comment
Line 98. Batch is better than family. Do barrels mean tanks? Please replace.
Response
Thank you for your carefully review. We have replaced “family” to “batch”, and barrels mean tanks, we are so sorry for our improper use of words.
Comment
Lines 103-107. It is not described how many replicates for each experimental condition and how many fish for each replicate.
Response
Thank you for your review. Our experimental group and control group both set up three replicates, with 5000 fish raised in each replicate tank. We have added in our revised manuscript.
Comment
Lines 105-107. Here the formula of the artificial feed is reported. But what are the constituents (raw materials) and their percentages? It would probably be more appropriate to report this information in a table.
Response
Thank you for your review. The formula ingredients for artificial feed include fish meal, krill meal, phospholipids, fish oil, soybean oil, flour, vitamins, mineral elements, etc. We have added the constituents in our revised manuscript. However, the specific percentage of these constituents has not been publicly disclosed by the producer. We are so sorry for not creating a table to report this information.
Comment
Line 101. “live bait fish”. More information is needed regarding the species, size, method of administration, density, method of distribution, etc. Have you evaluated the quality of live bait fish? Have you at least compared the energy balance between the two food theses?
Response
Thank you for your comment. The live bait fish we have used was Mrigal carp, with a size of approximately 80 fish per kilogram, and a feeding rate of 20% wet weight. That means We feed 2 kilograms of Mrigal carp to every 10 kilograms of mandarin fish. We have evaluated the quality of live bait fish. ISKNV and MRV tests were all negative. And we haven’t compared the energy between artificial feed and live bait fish. But we fed the Mandarin fish according to satiety. We found that the feeding amount of bait fish was four times that of artificial feed.
Comment
Lines 119-123. For commercial kits, the codes must be indicated.
Response
Thank you for your reminding. We have added the commercial kits’ codes in the revised manuscript.
Comment
Line 120. MDA is not an antioxidant assay.
Response
Thank you for your comment. MDA is not an antioxidant assay, but MDA is a biomarker that reflects the level of oxidative stress in the body, and its content can evaluate the level of tissue damage. So we put MDA content assay into the “Antioxidant Capacity Assays” section.
Comment
Line 153-150. According to my opinion, the correct statistical analysis is the two-way analysis of variance (ANOVA), with diet and days as factors, also considering their interaction.
Response
Thank you for your advice. We didn’t compare the variables over time because our fish are kept in factory captive tanks, there may be slight fluctuations in environmental factors such as dissolved oxygen and water temperature on different days. We believe that this comparison may not be able to achieve a single variable, so we did not compare them at different times. We used T-test to analyze the influence of diet to oxidative stress.
Comment
Lines 153-157 and Fig. 1. It is incorrect to represent growth as a weight gain but a standard growth rate (SGR). Moreover, in line 145 “The data are presented as mean ± standard deviation” is reported, whereas in Fig. 1 the data are presented as mean, SD, percentiles, and maximum and minimum values (maybe).
Response
Thanks for your suggestion. We added figure 1(C) in our revised manuscript. As shown in figure 3(C), there was no difference in the standard growth rate between domesticated Siniperca chuatsi fed with artificial feed and those fed with live bait in a short- term. And in figure 1, all the data are presented as mean ± standard deviation, the symbols in the figure are the repetition of every group.
Comment
Figs. 3 and 5. Other symbols appear in the bars; what do they represent, median, percentiles or what else?
Response
Thank you for your carefully comment. The symbols do not represent median, percentiles, they represent the different repetitive samples’ data. We apologize for the confusion caused to you. We have modified the style of the graphics to better observe the results.
Comment
The discussion needs to be reshaped. In some parts, information is indicated that is not always connected and discussed in light of the results obtained from this research.
Response
Thank you for your suggestion. We have reshaped the discussion part in our revised manuscript.
Comment
Line 262. Reference 44 is not reported.
Response
Thank you for your carefully check and reminder. Due to our carelessness, we missed a reference. We have made corrections in our revised manuscript.
Comment
The conclusions are similar to a short summary. They must be rewritten and mainly indicate what, in the light of the results obtained, are the prospects for breeding of this species and any new research on the merits.
Response
Thank you for your review. We have rewritten the summary section according to your suggestion in our revised manuscript.
Comment
Comments on the Quality of English Language. English needs a thorough overhaul. There are "non-sense" sentences and many grammatical or typo errors.
Response
Thanks for your comment. We apologize for some grammar errors caused by us not being native English speakers. We have applied for author service to correct the grammar and vocabulary errors in our article. Thank you very much for your suggestion.
Round 2
Reviewer 1 Report
Comments and Suggestions for Authors
No comments
Reviewer 2 Report
Comments and Suggestions for Authors
Deaar authors, I have only two minor suggestions for improving the manuscript. In the legend of figure 1, please insert that the symbols represent different repetitive samples' data. In the conclusions, delete "in summary" at the beginning of the paragraph.
Author Response
Thank you for review very much. We are very grateful for you reviewing our manuscript and providing the valuable comments.
comment
In the legend of figure 1, please insert that the symbols represent different repetitive samples' data.
Response
Thank you for your review. We have added the note that symbols “▲▼●■” represent different repetitive samples in our revised manuscript.
comment
In the conclusions, delete "in summary" at the beginning of the paragraph.
Response
Thank you for your suggestion. We have deletes “in summary” at the beginning of the paragraph in our revised manuscript.
comment
In the legend of figure 1, please insert that the symbols represent different repetitive samples' data.
Response
Thank you for your review. We have added the note that symbols “▲▼●■” represent different repetitive samples in our revised manuscript.
comment
In the conclusions, delete "in summary" at the beginning of the paragraph.
Response
Thank you for your suggestion. We have deleted “in summary” at the beginning of the paragraph in our revised manuscript.